# Copper(I) Complexes with Terphenyl-Substituted NPN Ligands Bearing Pyridyl Groups: Synthesis, Characterization, and Catalytic Studies in the S-Arylation of Thiols

**DOI:** 10.3390/molecules30153167

**Published:** 2025-07-29

**Authors:** M. Trinidad Martín, Ana Gálvez del Postigo, Práxedes Sánchez, Eleuterio Álvarez, Celia Maya, M. Carmen Nicasio, Riccardo Peloso

**Affiliations:** 1Instituto de Investigaciones Químicas (IIQ), Departamento de Química Inorgánica and Centro de Innovación en Química Avanzada (ORFEO−CINQA), Consejo Superior de Investigaciones Científicas (CSIC), Universidad de Sevilla, 41092 Sevilla, Spain; anaisabelgpg@gmail.com (A.G.d.P.); praxedes@iiq.csic.es (P.S.); ealvarez@iiq.csic.es (E.Á.); maya@us.es (C.M.); 2Departamento de Química Inorgánica, Universidad de Sevilla, 41071 Sevilla, Spain; mnicasio@us.es

**Keywords:** copper(I) complexes, phosphonites, NPN ligands, C-S catalytic coupling, single-crystal X-ray structures, diaminophosphines, terphenyl groups, pyridyl groups

## Abstract

In this study, three new terphenyl-substituted NPN ligands bearing pyridyl groups, two phosphonites and one diaminophosphine, were synthesized and fully characterized. Their coordination chemistry with copper(I) was investigated using CuBr and [Cu(NCMe)_4_]PF_6_ as metal precursors, affording six mononuclear Cu(I) complexes, which were characterized using NMR spectroscopy and, in selected cases, single-crystal X-ray diffraction (SCXRD) analysis. The NPN ligands adopt a κ^3^-coordination mode, stabilizing the copper centers in distorted tetrahedral geometries. The catalytic performance of these complexes in the S-arylation of thiols with aryl iodides was evaluated. Under optimized conditions, complexes **2a** and **2b** exhibited excellent activity and broad substrate scope, tolerating both electron-donating and electron-withdrawing groups, as well as sterically hindered and heteroaryl substrates. The methodology also proved effective for aliphatic thiols and demonstrated high chemoselectivity in the presence of potentially reactive functional groups. In contrast, aryl bromides and chlorides were poorly reactive under the same conditions. These findings highlight the potential of well-defined Cu(I)–NPN complexes as efficient and versatile precatalysts for C–S bond formation.

## 1. Introduction

Organic sulfur compounds constitute a broad class of naturally occurring substances with diverse biochemical and pharmacological properties, making them essential in various biological processes and medical applications [1,2,3]. In particular, thioethers are useful intermediates in the synthesis of a variety compounds that exhibit biological and pharmaceutical properties [4,5,6]. In recent decades, several non-catalytic synthetic strategies for the preparation of organic sulfides under mild conditions have been reported [7,8,9,10]. On the other hand, the development of catalytic processes for the synthesis of thioethers is generating increasing interest within the chemical community. In this regard, the metal-catalyzed coupling of organic halides with thiols is a straightforward way to obtain thioethers [11,12,13]. Although C-S coupling reactions have received less attention compared to other C-heteroatom bond formation reactions, mainly due to the fact that sulfur often causes catalyst poisoning and makes the reaction inefficient [14,15,16], considerable progress has been made in this field over the years [11,13,17,18,19,20].

Palladium-based catalysts for the S-arylation of thiols have been known since 1978 [19]. However, the high cost of this metal makes the development of processes based on more abundant and cheaper metals, such as copper, necessary. The first example of a copper-based catalytic system for C-S bond formation was reported by Palomo and coworkers in 2000 [21]. This system required a phosphazene-type base to carry out the couplings effectively. Since then, the use of copper(I) compounds as catalysts in the arylation of thiols has been actively developed, using more accessible bases and polydentate auxiliary N,O- or N,N-ligands [22,23,24,25]. Despite the advances, examples of the S-arylation of thiols catalyzed by well-defined copper species remain scarce.

In recent years, we have dedicated much effort to the synthesis of P(III) compounds functionalized with terphenyl groups, with the general objective of expanding this class of ligands and studying their coordination chemistry and catalytic properties in combination with late transition metals [26,27,28,29,30,31]. In particular, we successfully synthesized a family of dinuclear Cu(I) complexes of the general formula CuX(PR_2_Ar′) (X = Cl, Br, I; Ar′ = substituted *m*-terphenyl; R = alkyl, Figure 1), which performed remarkably as precatalysts in copper-catalyzed azide–alkyne cycloaddition reactions on water [29].

Herein, we report on the preparation and characterization of three new tridentate NPN-donor ligands of the phosphonite and diaminophosphine types, in which the nitrogen donors pertain two pyridyl rings. Moreover, we describe a family of Cu(I) complexes stabilized by the NPN ligands and their use as precatalysts in the S-arylation of thiols.

## 2. Results

### 2.1. Synthesis and Properties of Terphenyl-Substitued NPN Ligands Bearing Pyridyl Groups

On the basis of our recently reported procedures for the preparation of terphenyl-phosphonite ligands [27], we reacted the dihalophosphines PAr^Xyl2^X_2_ (X = Cl, Br; Ar^Xyl2^ = 2,6-bis (2,6-dimethylphenyl) phenyl) with the pyridine derivatives 2-hydroxypyridine, 2-hydroxy-6-methylpyridine, and 2-aminopyridine at an approximately 1.9:1 molar ratio in the presence of triethylamine to selectively produce the corresponding NPN ligands **NOPON^Xyl^^2^**, **NOPON^Xyl^^2^^-e^^2^**, and **N_2_PN_2_^Xyl^^2^**, respectively, as colorless solid materials, in good yields (Figure 2). Optimization of the reaction conditions implied the use of a slight excess of the parent dihalophosphines in order to avoid the presence of unreacted pyridines in the reaction mixture, which otherwise would require laborious work-up operations. Conversely, small amounts of residual PAr^Xyl2^X_2_ could easily be removed by washing the crude solid products with pentane.

The ^31^P{^1^H} NMR spectra of the three ligands in CDCl_3_ consiss of unique singlets in the 140–150 ppm range for the two phosphonites **NOPON^Xyl^^2^** and **NOPON^Xyl2^^-Me^^2^** and at a significantly lower frequency, ca. 38 ppm, for the diaminophosphine **N_2_PN_2_^Xyl^^2^**. As similarly observed for related terphenyl phosphonites, fast rotations around the P—C, P—O, and P—N bonds of the three molecules on the NMR time scale (Figure 1) account for the simplicity of their ^1^H NMR spectra, which can be rationalized by an apparent *C*_2v_ symmetry. Accordingly, for the three ligands, the four methyl groups of the xylyl rings give rise to one resonance at ca. 2 ppm, and the pyridyl groups originate only one pattern of signals. For the diaminophosphine ligand **N_2_PN_2_^Xyl^^2^**, a diagnostic broad peak at 5.04 ppm in CDCl_3_ solution is generated by the two equivalent protons of the NH fragments.

The diaminophosphine **N_2_PN_2_^Xyl^^2^** is indefinitely air-stable at room temperature, both as a pure solid material and dissolved in common organic solvents such as CHCl_3_, CH_2_Cl_2_, THF, and C_6_H_6_, even upon the addition of water. Contrarily, the phosphonites **NOPON^Xyl^^2^** and **NOPON^Xyl^^2^^-Me^^2^** have to be stored under an inert atmosphere of N_2_ or Ar to avoid hydrolysis yielding the corresponding phosphinic acid [27]. Thus, solid samples of pure **NOPON^Xyl^^2^** quantitatively converted into (O=)PH(OH)Ar^Xyl2^ (**1**) and 2-hydoxypyridine after 72 h of exposure to air (Figure 3). The same reaction products formed in approximately 24 h when pure samples of **NOPON^Xyl^^2^** were dissolved in THF/D_2_O or CDCl_3_/D_2_O mixtures. The presence of a P—H (or P—D) bond in the newly formed phosphinic acid **1** in CDCl_3_ solution, i.e., the predominance of the P(V) tautomer (O=)PH(OH)Ar^Xyl2^ over the P(III) one, P(OH)_2_Ar^Xyl2^, was doubly proven by the following: (a) a doublet centered at 6.93 ppm with a ^1^*J*_HP_ coupling constant of 575 Hz in the ^1^H NMR spectrum of samples of **1** produced without the addition of D_2_O; (b) a 1:1:1 triplet at 22.5 ppm (^1^*J*_PD_ = 117 Hz) in the ^31^P{^1^H} NMR spectrum samples of **1** obtained by reactions with D_2_O.

Slow evaporation of the solvent from CDCl_3_ solutions of **NOPON^Xyl^^2^,** prepared with the addition of one drop of D_2_O, allowed the isolation of a colorless crystalline material, which was identified by SCXRD analyses as the hydrogen-bonded adduct of the phosphinic acid **1** with 2-pyridone (Figure 2). The O–H···O interaction involves the hydroxyl hydrogen H2 of the phosphinic acid (donor, O2) and the carbonyl oxygen of the pyridone ring (acceptor, O3), with an O2···O3 (donor···acceptor) distance of 2.445(3) Å, an H2···O3 (H···acceptor) distance of 1.46(5) Å, and a corresponding angle of 169(5)°, consistent with a strong and highly directional hydrogen bond. An additional H-bond is observed between the nitrogen atom N1 of the pyridone and the phosphoryl oxygen O1 (N1···O1: 2.693(3) Å; O1···H1A: 1.81(4) Å; N1–H1A···O1: 178°). This structural arrangement is in line with similar hydrogen bond motifs observed in related phosphinic acid structures [32,33,34], all showing P(=O)OH···O = C hydrogen bonds with comparable geometrical parameters (O–H distances ≈ 0.82–0.95 Å, H···O distances ≈ 1.4–1.5 Å, and O···O distances ≈ 2.4–2.6 Å).

Presumably as a consequence of the additional steric protection provided by the two methyl groups of the 6-methylpyridyl substituents, the decomposition of **NOPON^Xyl2^^-Me^^2^** caused by water is significantly slower compared to that of **NOPON^Xyl^^2^**. Using the same reaction conditions employed for **NOPON^Xyl^^2^**, the hydrolysis of **NOPON^Xyl^^2^^-Me^^2^** took roughly twice the time.

### 2.2. Synthesis and Characterization of Cu(I) Complexes with ***NOPON^Xyl^^2^***, ***NOPON^Xyl^^2^^-^^Me^^2^***, and ***N_2_PN_2_^Xyl^^2^***

Two commonly used copper(I) precursors, CuBr and [Cu(NCMe)_4_]PF_6_, were reacted with the three NPN ligands described in the previous section to prepare two series of Cu(I) complexes of the general formulas CuBrL (L = **NOPON^Xyl^^2^**: **2a**; **NOPON^Xyl^^2^^-Me^^2^**: **2b**; **N_2_PN_2_^Xyl^^2^**: **2c**) and [Cu(NCMe)L]PF_6_ (L = **NOPON^Xyl^^2^**: **3a**; **NOPON^Xyl^^2^^-Me^^2^**: **3b**; **N_2_PN_2_^Xyl^^2^**: **3c**), respectively (Figure 4).

The formation of the aforementioned complexes was confirmed by their ^31^P{^1^H} NMR spectra in the CDCl_3_ solution, which exhibited broad singlets up-field shifted by approximately 20–30 ppm relative to the free ligands. ^1^H NMR spectra of complexes **2a**–**c** do not show relevant differences compared to those of the ligands, whereas the presence of a coordinated acetonitrile molecule in complexes **3a**–**c** is evidenced by a singlet at ca. 2.4–2.5 ppm corresponding to the CH_3_ group.

The slow diffusion of petroleum ether into dichloromethane solutions of complexes **2b**, **2c**, and **3b** at approximately −20 °C yielded single crystals suitable for SCXRD analysis, enabling the determination of their molecular structures in the solid state. The corresponding ORTEP representations are shown in Figure 3, Figure 4 and Figure 5, respectively, along with the corresponding lists of selected bond distances and angles. In all three complexes, the NPN ligands adopt a κ^3^ coordination mode, with the metal center lying in a markedly distorted tetrahedral environment, resembling a disphenoidal or seesaw-like arrangement. The bond angles at copper range from ca. 80–85°, corresponding to the P-Cu-N angles within the cupracycles, to ca. 130–150° for the P-Cu-Br and P-Cu-NCMe angles. These angular distortions are reflected in the values of the four-coordinate geometry index, τ_4_ [35], which are 0.68, 0.69, and 0.67 for complexes **2b**, **2c**, and **3b**, respectively. Notably, as observed in the right-side views shown in Figure 3, Figure 4 and Figure 5, the cupracycles in complex **2c** are nearly perfectly planar, whereas those in **2b** and **3b** appear slightly twisted, likely due to the increased steric hindrance introduced by the methyl substituent on the pyridyl rings.

Terphenyl groups have shown the ability to stabilize unsaturated coordination compounds by means of metal–carbon secondary interactions [26,27,28,29,30,31]. Taking this into account, we reacted complex **2b** with AgBF_4_ in dichloromethane, aiming to replace the Cu-Br bond with a weaker non-covalent interaction involving the terphenyl ring. The ^31^P{^1^H} NMR spectrum of the crude material obtained after filtration and evaporation of the volatiles revealed the formation of two new species associated with singlet resonances at 23.5 ppm and 25.5 ppm. The slow evaporation of CDCl_3_ from the NMR tube yielded pale yellow crystals of the dinuclear complex [Cu_2_Cl(N_2_PN_2_^Xyl2^)_2_]BF_4_, **4**, in which two [Cu(N_2_PN_2_^Xyl2^)]^+^ units are bridged by a chloride anion (Figure 6). Although no mechanistic studies were performed, the formation of compound **4** can likely be attributed to the use of dichloromethane as the reaction medium together with the high reactivity of the in situ-generated [Cu(N_2_PN_2_^Xyl2^)]^+^ cation. Furthermore, the slight difference observed between the two ^31^P{^1^H} NMR signals in the initial mixture suggests that the second product is plausibly the bromine-bridged analog of **4**.

### 2.3. Catalytic Studies of the S-Arylation of Thiols

To test the catalytic activity of complexes **2a**–**2c** in the arylation of thiols, the reaction of thiophenol with iodobenzene was chosen as the model system. Catalytic experiments were initially performed using the conditions previously applied for copper-assisted C-S bond formation, namely, dioxane solution at 110 °C, NaO*t*Bu as the base, and 5 mol% catalyst loading [23]. Under these conditions, nearly quantitative yields of the C-S coupling product, diphenyl sulfide, were obtained using **2a** as the catalyst system (Table 1, entry 1). Since the base NaO*t*Bu is incompatible with several functional groups, such as esters or aldehydes, other bases were tested. Notably, the weaker inorganic base K_3_PO_4_ provided full conversion to the thioether product (entry 3). Lowering the temperature to 100 °C produced a slight decrease on the reaction yield (entry 4). At this temperature, the use of other solvents did not improve the reaction outcome (entries 5 and 6).

However, reducing the catalyst loading down to 2 mol% did not produce an appreciable decrease in the catalytic efficiency of the system, although lower loadings significantly lessened the reaction yield (entries 7–9). Under the optimized reaction conditions, complex **2b** performed very similarly to **2a**, whereas complex **2c** showed significantly lower activity, possibly due to the deprotonation of the diaminophosphine **N_2_PN_2_^Xyl2^** ligand under basic conditions. Finally, no reaction was observed in the absence of a copper catalyst.

Encouraged by the results obtained using iodobenzene, we tested bromobenzene and chlorobenzene as electrophiles in the S-arylation of thiophenol (Table 2).

In all experiments conducted with **2a** as the catalyst, very low conversions were obtained, even when increasing the catalyst loading up to 10 mol% or using different solvents and bases.

Complexes **2a** and **2b** were both chosen to explore the scope of C–S coupling reactions under the optimized reaction conditions. As summarized in Figure 5, a broad range of (hetero)aryl iodides were coupled with a variety of thiophenols, affording the corresponding thioethers in good-to-excellent yields. Both electron-donating (**b**–**g**, **k**–**o**, **q**) and electron-withdrawing substituents (**h**–**i**) on the aryl iodides were well tolerated. Moreover, the increased steric demand due to the *ortho*-substitution on both coupling partners did not require any modification of the reaction conditions, including catalyst loading (**e** and **f**). Notably, aryl halides such as 1-chloro-4-iodobenzene and 1-bromo-4-iodobenzene underwent selective coupling with thiophenols exclusively at the C-I bond, while C-Cl and C-Br bonds remained unaltered (**h** and **i**). Furthermore, nitrogen-containing heteroaryl halides served as effective electrophilic coupling partners, delivering the desired products in high yields (**j**, **q,** and **r**).

Regarding the scope of the nucleophile, *p*-substituted thiophenols with both electron-withdrawing (**l**) or electron-donating groups (**f**, **g**, **m**, and **n**) exhibited comparable reactivity. Importantly, thiols featuring functional groups prone to arylation, such as unprotected hydroxyl or amine groups, underwent C–S coupling with complete chemoselectivity (**m** and **n**), eliminating the need for protective groups. Furthermore, the coupling with aliphatic thiols, which are typically more challenging due to their lower nucleophilicity, was also explored under the same reaction conditions. Primary (**r**), secondary (**o** and **p**), and tertiary alkyl thiols (**q**) were efficiently coupled with a variety of (hetero)aryl iodides, demonstrating the broad applicability of this catalytic protocol.

## 3. Materials and Methods

All synthetic procedures and manipulations were performed under an oxygen-free nitrogen atmosphere using standard Schlenk techniques. Solvents were thoroughly dried and degassed prior to use. Copper(I) bromide was obtained as a colorless solid via the aqueous reduction of CuSO_4_·5H_2_O (1 equiv) with Na_2_SO_3_ (2 equiv) in the presence of sodium bromide (approx. 4 equiv). The resulting solid was isolated by filtration, washed with acetic acid, and diethyl ether, dried under a vacuum, and stored under nitrogen. The dihalophosphines PX_2_Ar^Xyl2^ [36] and [Cu(MeCN)_4_]PF_6_ [37] were synthesized following procedures from the literature. All other reagents were purchased commercially and used without further purification.

Solution NMR spectra were recorded on Bruker Avance DPX-300 and DPX-400 spectrometers. Chemical shifts for ^1^H and ^13^C NMR were referenced to residual solvent peaks, while ^31^P NMR shifts were referenced externally to H_3_PO_4_. Elemental analyses were carried out by the Microanalysis Service at the Instituto de Investigaciones Químicas (IIQ). Single-crystal X-ray diffraction (SCXRD) studies were conducted at the Centro de Investigación, Tecnología e Innovación de la Universidad de Sevilla (CITIUS), and at the Instituto de Investigaciones Químicas, cicCartuja (Seville). Detailed synthetic procedures for all ligands and metal complexes, along with their spectroscopic data, are provided in the Appendix A.

### General Catalytic Procedure for the S-Arylation of Thiols with Aryl Iodides

Solid samples of **1** or **2** (0.02 mmol) were dissolved in dioxane (1 mL). The aryl iodide (1.2 mmol), the thiol (1.0 mmol), and K_3_PO_4_ (2.0 mmol) were added under a nitrogen atmosphere. After 24 h at 110 °C, the reaction mixture was allowed to cool to room temperature, diluted with ethyl acetate (10 mL), and filtered through Celite. The conversion was determined via GC analysis. Pure products were obtained after purification via flash chromatography on silica gel with petroleum ether (unless otherwise indicated) and identified using ^1^H NMR spectroscopy for a comparison with literature data [38,39,40,41,42,43,44,45,46,47,48,49].

## 4. Conclusions

Three novel terphenyl-substituted NPN ligands bearing pyridyl groups were synthesized from the parent dihalophosphines in high yields. Their coordination to copper(I) precursors CuBr and [Cu(NCMe)_4_]PF_6_ led to the formation of six mononuclear Cu(I) complexes, in which the ligands adopt a κ^3^-coordination mode, stabilizing the copper centers in distorted tetrahedral geometries. Complexes **2a** and **2b** demonstrated high catalytic activity in the S-arylation of thiols with aryl iodides, providing the efficient coupling of an ample range of (hetero)aryl iodides with both aromatic and aliphatic thiols. In addition, the catalytic protocol showed excellent chemoselectivity and functional group tolerance. These findings highlight the potential of well-defined Cu(I)–NPN complexes as efficient and versatile precatalysts for C–S bond formation, offering a promising alternative to more expensive or toxic metal-based systems.

## Data Availability

The original contributions presented in this study are included in the article. Further inquiries can be directed to the corresponding authors.

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
