# Peer review of "Copper(I) Complexes with Terphenyl-Substituted NPN Ligands Bearing Pyridyl Groups: Synthesis, Characterization, and Catalytic Studies in the S-Arylation of Thiols"

_molecules, 2025, doi:10.3390/molecules30153167_

Round 1

Reviewer 1 Report

Comments and Suggestions for Authors

As coordination chemist working occasionally also on the coordination of thioether ligands on metal complexes I have read this sound manuscript on the design of novel NPN ligands, their coordination on Cu(I) salts and the catalytic activity for thiol coupling reactions with many interests. The content of this contribution matches in my opinion well with the topic of the special issue  “Inorganic Chemistry in Europe 2025” . The amount of work is sufficient; the scholar presentation is satisfying, and all compounds are well characterized, and useful additional information is provided in the supporting material. Also the content is of sufficient originality and novelty.

I have just some minor remarks:

- Why CuI has not been used as starting material?

-I have some doubts about the toxicity of Pd as stated by the authors in line 44. This is the most used metal employed

The authors should somewhat more discuss the hydrogen bonding occurring in crystal structure shown in Fig. 2

-Where are the checkcifs? Without them its difficult to comment on the quality of an X-ray structure. The submission of checkcifs should be mandatory.

- The authors should at least suggest how the Cu(I) complexes are involved in the catalytic reaction (catalytic cycle)

Author Response

1 - Why CuI has not been used as starting material?

CuBr was chosen as the metal precursor because it was readily available in the laboratory. CuI was employed by some of us in the synthesis of various Cu(I) complexes featuring bulky phosphines, including terphenyl phosphines (Inorg. Chim. Acta 2023, 556, 121623; Inorg. Chem. 2020, 59, 15, 10894–10906). In all cases, we observed the formation of neutral dinuclear species, regardless of the nature of the bridging halide. In other words, changing the halide did not significantly affect the structural features of the Cu(I) complexes studied, except for variations in the Cu–X–Cu bond angles and Cu···Cu distances. Given the chelating nature of the ligand used in the present study, we anticipated that the halide would not substantially influence the properties of the resulting complexes. Therefore, we opted to prepare cationic complexes using Cu(MeCN)₄PF₆.

2- I have some doubts about the toxicity of Pd as stated by the authors in line 44. This is the most used metal employed.

We thank the referee for this comment. Actually, the toxicity of Pd might be similar to those of Ni or Cu, depending of the type of compund and the life being that is exposed to it (Angew. Chem. Int. 2016, 55, 12150). We changed the sentence, as follows: "However, the high cost of this metal, make necessary the development of processes based on more abundant and cheaper metals, such as copper". 

3 - The authors should somewhat more discuss the hydrogen bonding occurring in crystal structure shown in Fig. 2.

Accordingly, we have expanded the discussion of the hydrogen bonding interactions in the solid-state structure of the adduct shown in Figure 2. The new manuscript paragraph in Section 2.1 now includes a more detailed analysis of the O–H···O and O···N contacts.

4- Where are the checkcifs? Without them its difficult to comment on the quality of an X-ray structure. The submission of checkcifs should be mandatory.

They have been included in the Supplmentaty Material.

- The authors should at least suggest how the Cu(I) complexes are involved in the catalytic reaction (catalytic cycle).

Unlike cross-coupling reactions catalyzed by Pd complexes, there is not a unique mechanism for Cu-catalyzed Ullmann type cross-coupling reactions. Both Cu(I)/Cu(III) and Cu(I)/Cu(II) catalytic manifold have been proposed (see new references 24 and 25). Since we have not carried out any mechanistic studies with our Cu(I) complexes, we are not confident in proposing a mechanism for the transformations outlined in this work.  

Reviewer 2 Report

Comments and Suggestions for Authors

Martín et al. present a very nice study on the design, synthesis, and catalytic application of novel terphenyl-substituted NPN ligands bearing pyridyl groups, with particular focus on their coordination behavior toward copper(I) and subsequent performance in C–S coupling reactions. The work successfully demonstrates highly active and chemoselective precatalysts for the S-arylation of thiols with aryl iodides. The authors have characterized through detailed spectroscopic and crystallographic characterization the complexes, alongside a broad and well-optimized catalytic scope. The novelty of this work lies in the rational ligand design and structural control over the metal coordination environment, which enables high functional group tolerance and remarkable chemoselectivity. I think that the manuscript presented is well-presented, and of clear interest to a broad readership in organometallic and catalytic chemistry. I suggest its publication in Molecules after minor revision:

  • Line 31 page 1: “Organic sulfur compounds constitute a large family of naturally occurring com-30 pounds with a variety of unusual properties [1-3]” The authors should justify the statement “unusual”.
  • Line 54 page 2: “In particular, we successfully synthesized a 54 family of dinuclear Cu(I) complexes of the general formula CuX(PR2Ar′) (X = Cl, Br, I; Ar’ 55 = substituted m-terphenyl; R = alkyl)” The authors could help the reader by introducing a sketch of the previous complexes studied.
  • The authors should report a detailed crystallization procedure of the different compounds solved by SXRD.

Typos/grammatical

Line 37 page 1 “thiothers” should be thioethers

Line 87 pag 3 “generetatated should be generated

Line 203 page 7 “produced a slightly decrease” should be slight decrease

Line 208 page 7 “S-arylation of tiophenol” should be thiophenol

Line 145 pag 4 “These angular distorsions are reflected” should be distortions

Author Response

1 - “Organic sulfur compounds constitute a large family of naturally occurring com-30 pounds with a variety of unusual properties [1-3]” The authors should justify the statement “unusual”.

We changed the sentence as follows: "Organic sulfur compounds constitute a broad class of naturally occurring substances with diverse biochemical and pharmacological properties, making them essential in various biological processes and medical applications". 

2-  “In particular, we successfully synthesized a 54 family of dinuclear Cu(I) complexes of the general formula CuX(PR2Ar′) (X = Cl, Br, I; Ar’ 55 = substituted m-terphenyl; R = alkyl)” The authors could help the reader by introducing a sketch of the previous complexes studied.

A scheme has been introduced (Scheme 1).

3 - The authors should report a detailed crystallization procedure of the different compounds solved by SXRD.

We thank the referee for this comment. This phrase has been added at page 5: "Slow diffusion of petroleum ether into dichloromethane solutions of complexes 2b, 2c, and 3b at approximately –20 °C yielded single crystals suitable for SCXRD analysis, enabling the determination of their molecular structures in the solid state".

Typos have been corrected.